# Household latrine utilization and associated factors in semi-urban areas of northeastern Ethiopia

Deres Asnake[1]☯, Metadel Adane[2]☯*

1 Quality Improvement Unit, Woldia Comprehensive Specialized Hospital, Woldia, Ethiopia, 2 Department of Environmental Health, College of Medicine and Health Sciences, Wollo University, Dessie, Ethiopia

☯ These authors contributed equally to this work.
* metadel.adane2@gmail.com

## Abstract

### Background

Latrine utilization is the actual behavior in a practice of regularly using existing latrines for safe disposal of excreta. Latrine utilization is a common problem in semi-urban areas of developing countries, including Ethiopia. Since the status of latrine utilization and associated factors among semi-urban areas of northeastern Ethiopia, including Alansha in South Wollo Zone is unknown, local data is needed in order to assess the need for planning of intervention programs for the improvement of latrine utilization to support consistent and sustained latrine utilization. This study is designed to address this knowledge gap.

### Methods

A cross-sectional study was conducted from February to March, 2019 among 401 systematically selected households. Data were collected by trained workers using a pre-tested, structured questionnaire via face-to-face interviews and on-the-spot observations of the latrines. A systematic random sampling method was used to select participant households. Data were entered using EpiData version 3.1 and exported to Statistical Package for the Social Sciences (SPSS) version 25.0 for data cleaning and analysis. The wealth index status of participants was estimated using principal component analysis. Data were analyzed using a binary logistic regression model at 95% confidence interval (CI). From the multivariable logistic regression analysis, variables with $p$-value < 0.05 were taken as statistically significant and independently associated with latrine utilization. Model fitness was checked using Hosmer-Lemeshow test.

### Result

We found that the prevalence of latrine utilization among households was 71.8% (95% CI [67.5–76.1%]) while 28.2% (95% CI [23.9–32.5%]) did not utilize latrines. About one-fifth (21.7%) of participant households were found to have a pit latrine with slab and 78.3% (311) used pit latrines without slab. The hygienic condition of the majority (82.9%) of the latrines

**Data Availability Statement:** All relevant data are within the paper and its Supporting Information file.

**Funding:** Amhara Regional Health Bureau (ARHB) funded this study. The funders had no role in study

design, data collection and analysis, decision to publish, or preparation of the manuscript.

**Competing interests:** The authors have declared that no competing interests exist.

**Abbreviations:** AOR, adjusted odds ratio; CI, confidence interval; COR, crude odds ratio; CI, confidence interval; EDHS, Ethiopia Demographic and Health Survey; WASH, water, sanitation and hygiene.

was dirty and only 17.1% clean. Household family size from one to three persons (AOR: 3.99, 95% CI [1.20–6.24]), presence of primary or secondary school student in a house (AOR: 2.33, 95% CI [1.42–3.83]), number of years since latrine was constructed ($\geq$ 2 years) (AOR: 1.82, 95% CI [1.12–2.95]) and a frequency of daily cleaning of the latrine (AOR: 2.19, 95% CI [1.12–4.28]) were factors significantly associated with latrine utilization.

## Conclusion

Seven out of ten households utilized a latrine. Factors significantly associated with latrine utilization were household family size from one to three persons, presence of primary or secondary school student in the house, time since household latrine had been constructed of two or more years and daily frequency of latrine cleaning. Therefore, it is recommended that measures to promote behavioral change towards further improvement in sustainable and consistent latrine utilization should be carried out based on the evidence of the determinant factors found in this study.

## Background

A lack of sanitation facilities compels people to practice open defecation. In 2010, 15% of the world's population still practiced open defecation [1]. Open defecation contributed to poor household sanitation, which increases the risk of transmission of diseases such as diarrhea [2]. Globally, 2.3 billion human beings still do not have access to basic sanitation facilities, of whom the highest proportions are found in South Asia and Sub-Saharan Africa [3]. In Africa, the presence of open defecation has been attributed to a likely cultural-habitual preference for open defecation practice and inadequate water availability [4]. Pit latrines are in use by more than half the urban population in Sub-Saharan Africa (SSA); however, the performance of many of these types of latrines have been found to be unsatisfactory [5]. A worldwide systematic review and meta-analysis study found that latrine use was associated with better maintenance, accessibility, privacy, facility type, cleanliness, and better hygiene access [6].

The 2016 Ethiopia Demographic and Health Survey (EDHS) showed that the extent of national open defecation was 32.9% and more than half (56%) of rural households used unimproved toilet facilities [7]. In Ethiopia, latrine facility coverage has been increasing since the health extension [8, 9] and water, sanitation and hygiene (WASH) program started [10]. As a result, the rate of reduction of open defecation has been remarkable; a 25% reduction per decade for the years 1990–2015 (from 92% in 1990 to 29% in 2015) [3]. However, there has been comparably less attention to utilization of latrine facilities in rural areas versus urban areas of Ethiopia [11].

Problems related to latrine utilization include lack of a functional latrine, latrines located a long distance away from areas where farmers work, and latrines lacking a superstructure [12]. Another study in rural Ethiopia showed that latrine use was due to ownership of a latrine that had a superstructure, having a clean latrine, and having a latrine with a protected door [13]. A study conducted in the Amhara region in Ethiopia found that latrine utilization was associated with education, relative wealth, urban residence and history of travel [14]. Furthermore, a study in Denbia District, found that integration of strategies to promote hygiene behavioral change with construction of sanitation facilities is crucial; and that a long-established habit

and comfort with open defecation was the main reason of 60.4% of adults for not using a latrine [11].

These gaps point out the need for this study in order to establish evidence-based information on latrine utilization status and to identify factors associated with the utilization of latrine in the previously-unstudied areas of semi-urban Alansha, Ethiopia. An understanding of latrine utilization in this area will guide the planning of targeted intervention programs for improvement of household latrine utilization, which is included in the United Nations Sustainable Development Goals (SDGs), particularly in Goal 6, which includes the target of achieving access to basic sanitation for all [15]. The findings of this study may help to improve latrine utilization in the development of improved sanitation facilities in the semi-urban areas in Alansha in northeastern Ethiopia and throughout semi-urban areas in Ethiopia.

## Methods

### Study design and study area

A cross-sectional study was conducted from February to March, 2019 in the semi-urban areas of Alansha, located in Kutaber District of South Wollo Zone in northeastern Ethiopia. Alansha had one semi-urban *kebel*e. *Kebele* is the smallest administrative unit in Ethiopia, with an average population of 5,000. Agriculture is the area's main economic activity. It has a total population of 8,907 including 4,426 (49.7%) men and 4,481 (50.3%) women, and a total of 1,845 households. Two elementary schools and one health post are found in the area. Alansha is situated on a plateau and some sloping areas in a mountainous region that runs north to south. Latrine coverage in the areas is 99% (Semi-urban Alansha *kebele* Administration, unpublished document data, 2018).

### Sample size determination and sampling procedures

The sample size was determined using the single population proportion formula [16] considering the assumptions of:

$$n = \frac{\left(z_{(a/2)}\right)^2 * p(1-p)}{d^2}$$

$Z_{\alpha/2}$ at 95% confidence interval (CI) is 1.96, $p$ is an estimate of the proportion of latrine utilization (57.3%), which is taken from a similar study conducted in Tigray Region, Gulomekada District, Ethiopia [17] and $d$ margin of error (5%). A sample size correction formula was also employed since the source population (8,907) was less than 10,000. After adjusting for an anticipated 10% non-response rate, the final sample size was determined to be 401.

The semi-urban area of Alansha had one *kebel*e with 1,845 total households, including a total population 8,907 from which a final sample size of 401 was included. The source population of this study was all people in the semi-urban area of Alansha and the study population was those households selected from residents of semi-urban Alansha. Systematic random sampling was used to find households; total households divided by final sample size gave a sampling interval of 5. Then, data were collected at an interval of every fifth house.

### Inclusion and exclusion criteria

All households with a functional latrine during the study period were included. Members of each household who were less than 18 years old during the data collection period were excluded as study participants.

## Operational definitions

**Semi-urban.** A partially urban, partially rural area that was not within a municipality; where the population of city residents was 2,000 or more; where it was estimated that there was potential to earn income, on average, other than land lease sale, of $896.50 USD (1 USD [United States Dollars] = 27.8862 Ethiopia birr during the study period) or more per year; where 20% of city residents engaged in an occupation other than agriculture; that may serve as a development center; that had good transportation [18].

**Latrine utilization.** Determined using "signs of use" such as a household having a functional latrine, children's faeces being safely disposed of, no observable faeces in the compound and at least one observable sign of use (e.g., foot path to the latrine not covered by grass, latrine odor, lack of spider web in squatting hole, presence of anal cleansing material, fresh faeces in the squatting hole, or a wet slab) [19].

**Safe disposal of child faeces.** Children's faeces disposed of in a toilet, not in the open.

**Poor latrine.** A latrine without superstructure and lacking walls.

**Fair latrine.** A latrine having superstructure, without a door (any cover) but with a leaking roof at time of data collection.

**Good latrine.** A latrine having superstructure, with a door (any cover) and possibility of maintaining privacy during defecation.

**Dirty latrine.** Visible faeces and/or urine on the floor around the latrine and latrine not swept at the time of data collection.

**Clean latrine.** Pit not full, no faecal matter seen around the pit latrine, area properly swept and absence of bad smell at time of data collection.

**Open defecation.** Self-reported behavior, including defecating in fields, bushes, forests, open bodies of water, or other open spaces [20].

**Sanitation.** The provision of facilities for the safe disposal of human faeces and urine [21].

## Variables measured

The outcome variable of this study was latrine utilization, which is a binary outcome denoted as yes (1) for latrine utilized or no (0) for latrine not utilized. The independent variables included socio-demographic and economic, environmental and behavioral factors. Socio-demographic variables that were considered in this study as potential confounders and measured by face-to-face interviewers included head of household marital status, occupation, educational status, sex, age, religion; presence of primary or secondary school student in the household; presence of children under five; and household size (number of persons). The wealth index of a household was also computed using principal component analysis. A wealth index score was classified using the EDHS 2016 five categories as lowest, second, middle, fourth and highest income [7].

Latrine-related variables that were measured by on-the-spot observation were type of latrine, condition of latrine (poor, fair, good), faeces seen around pit hole/floor of latrine, latrine location, presence of squat hole cover, latrine slab sealed with mud/cement and presence of latrine walls, roof and door. Latrine variables that were measured in self-reported of study participants included the means of disposal of under-five children's faeces, number of years since the latrine had been constructed and number of times a latrine had been constructed. Distance of latrine from the house was measured in meters.

Behavioral variables that were measured by self-report of the study participants included frequency of latrine cleaning, latrine hygienic condition, whether information had been received about constructing a latrine, reasons for constructing a latrine, what person was responsible for constructing latrine, whether a lack of latrine was considered as culturally

taboo and types of taboos. Presence of handwashing facility in/near latrines was measured by on-the-spot observation.

## Data collection and data quality assurance

Data were collected by face-to-face interviews and on-the-spot- observation. The questionnaires were first prepared in English and translated to Amharic, and then re-translated back to English to ensure consistency. A pretest of the questionnaire was conducted with 5% from the total sample size (20 of 401 questionnaires) to check the consistency and clarity of the questions. It was undertaken on 20 households that were not included in the study before the actual data collection period. The aim was to determine if there were any difficulties filling out the questionnaire, challenges in interviewing, or misunderstanding of the questions by enumerators. During data collection, one supervisor collected the questionnaires from each enumerator on a daily basis, checking the consistency and the completeness of the completed questionnaire on the spot.

The four data collectors were environmental health professionals with a BSc degree. The principal investigator provided one day of training for the four data collectors and one supervisor before actual data collection took place. The training was focused on how to fill out the questionnaire through interview and on-the-spot observation of the latrines, how to approach the study participants and about ethical issues during the data collection. Field supervision and daily meetings were conducted to solve any problems that came up to ensure the quality of data collection. To check the quality of the entered data, 10% of the entered questionnaires were randomly selected and re-entered to control data entry errors.

## Data management and analysis

All field questionnaires were first checked, coded and entered in to EpiData version 3.1 (EpiData Association, Odense, Denmark) statistical software and then exported into Statistical Package for the Social Sciences (SPSS) version 25.0 (IBM Corp., Armonk, N.Y., USA) for data cleaning and analysis. Principal component analysis was used to construct the household wealth index with the considerations of the assumptions: communality value > 0.5, Kaiser-Meyer-Olkin (KMO) value > 0.5, and eigenvalues greater than one [22]. Wealth index was calculated by national wealth quintiles compiled by assigning the household score to each usual (*de jure*) household member, ranking each person in the household population by her or his score, and then dividing the distribution into five equal categories, namely lowest, second, middle, fourth and highest. The prevalence of latrine utilization was estimated from the proportion of individuals who practiced proper latrine utilization within the total number of study households (total sample size) with functional latrine was multiplied by 100.

The presence of multi-collinearity among independent variables was checked using standard error at the cutoff value of 2 [23], which was not observed. Binary logistic regression model was fitted to assess factors associated with latrine utilization. Bivariate analysis (crude odds ratio [COR]) with 95% CI was used to assess the crude association and to select important variables to be included in the final model. From the bivariate analysis, $p<0.25$ was retained into multivariable logistic regression model analysis. Finally, multivariable-logistic regressions (adjusted odds ratio [AOR]) with 95% CI was used to control potential confounders and to identify independent predictors of latrine utilization. From the adjusted analysis, a significance level of $p < 0.05$ was declared as a factor significantly associated with latrine utilization. Model fitness was checked using Hosmer-Lemeshow test [23] and the model was fit at $p$-value = 0.957.

### Ethical approval and consent to participate

An ethical clearance letter was obtained from the Ethical Review Committee of Wollo University College of Medicine and Health Sciences (protocol number: WU/ERC/501/02/19). All study participants were informed about the purpose of the study, and their verbal consent was obtained and recorded by data collectors. The ethics committee approved the verbal consent procedure because the study interviews about latrine utilization and spot-check observation of latrines were not considered to have major ethical issues, and no blood or other ethically sensitive samples were taken. In the study area, latrine observation and interviews by local health experts are common and routine activities for the study participants. The respondents' right to refuse or withdraw from participating the interview was fully maintained and the information provided by each respondent was kept strictly confidential.

## Results

### Socio-demographic and economic characteristics of the study participants

In this study, of the total 401 participants, 397 responded, for a response rate of 99.0%. More than two-third 69.3% (275) of participants were female and the rest were male 30.7% (122). The number of people who were illiterate accounted for half 51.6% (205) of study participants. The occupation of more than one-third 41.8% (166) of the study participants was housewife, whereas 36.5% (145) were farmers. The household size of a majority of the households was one to three persons 91.9% (365) and three-fourths 75.1% (298) of the households had a primary or secondary school student in the house. One-fourth 26.2% (104) of households were in the lowest wealth index category, whereas one-fifth 20.2% (80) were in the middle income category and about one-tenth 10.8% (43) were in the highest income category (Table 1).

From the bi-variable analysis of socio-demographic factors, households having a wealth index in the second category had 1.44 times (COR: 1.44, 95% CI [0.80–2.57]) higher latrine utilization than those households in the lowest wealth index category. The odds of latrine utilization among households that had a family size of one to three persons were 3.8 times (COR: 3.81, 95% CI [1.18–7.30]) higher than among households that had a family size of greater than six persons. Households that included a primary or secondary school student were 2.4 times (COR: 2.40, 95% CI [1.49–3.88]) more likely to utilize a latrine than households that did not (Table 1).

### Latrine characteristics

Among households included in the study, 21.7% (86) were found to use a pit latrine with slab and 78.3% (311) used pit latrines without slab. The time since the latrine had been constructed was two years or more for 43.8% (174) of households. About 31.2% (124) of latrines were located inside the house compound and 48.6% (193) outside the house compound. Nearly half 47.1% (187) of households had constructed latrines for the second time and about one-fourth 23.4% (93) households had constructed latrines for the first time. About three-fourth 74.1% (294) of latrines had a door and 25.9% (103) did not have a door. The condition of half 49.6% (197) of the latrines was poor and almost one-tenth 12.1% (48) of latrines were in good condition. A majority 78.3% (311) of the latrines slab were not sealed with mud/cement (Table 2).

From the bivariate analysis, we found that the odds of a pit latrine with slab being utilized was 1.39 times (COR: 1.39, 95% CI [0.80–2.42]) higher than a pit latrine without slab. The odds of a latrine being utilized by households whose latrine had been constructed two or more years previously were 1.78 times (COR: 1.78, 95% CI [1.23–2.80]) greater than those households whose latrines had been constructed less than two years previously. Study participants

**Table 1. Socio-demographic characteristics and bivariate analysis with latrine utilization in semi-urban areas in northeastern Ethiopia, February to March, 2019.**

| Variables | Latrine utilization (N = 397) | | | COR (95%CI) | p-value |
|---|---|---|---|---|---|
| | Frequency | Utilized (n = 285) | Not utilized (n = 112) | | |
| | %(n) | %(n) | %(n) | | |
| **Head of household** | | | | | |
| Mother | 78.3(311) | 21.8(62) | 21.4(24) | 1.02(0.59–1.73) | 0.943 |
| Father | 21.7(86) | 78.2(223) | 78.6(88) | Ref | |
| **Sex** | | | | | |
| Male | 30.7(122) | 31.9(91) | 27.7(31) | 1.23(0.76–1.99) | 0.409 |
| Female | 69.3(275) | 68.1(194) | 72.3(81) | Ref | |
| **Age (years)** | | | | | |
| 18–40 | 25.2(100) | 23.5(67) | 29.5(33) | Ref | |
| 41–60 | 41.1(163) | 43.2(123) | 35.7(40) | 1.51(0.87–2.62) | 0.138* |
| 61–80 | 19.1(76) | 18.2(52) | 21.4(24) | 1.06(0.56–2.02) | 0.842 |
| >81 | 14.6(58) | 15.1(43) | 13.4(15) | 1.4(0.68–2.9) | 0.348 |
| **Religion** | | | | | |
| Muslim | 95.5(379) | 95.8(273) | 94.6(106) | 1.29(0.47–3.52) | 0.622 |
| Christian | 18(4.5) | 4.2(12) | 5.4(6) | Ref | |
| **Occupation** | | | | | |
| Housewife | 41.8(166) | 44.2(126) | 35.7(40) | 1.32(0.80–2.20) | 0.270 |
| Day laborer | 8.1(32) | 6.3(18) | 12.5(14) | 0.54(0.25–1.19) | 0.126* |
| Government employee | 6.3(25) | 5.6(16) | 8.0(9) | 0.75(0.31–1.83) | 0.526 |
| Merchant | 7.3(29) | 8.1(23) | 5.4(6) | 1.62(0.62–4.25) | 0.330 |
| Farmer | 36.5(145) | 35.8(102) | 38.4(43) | Ref | |
| **Educational status** | | | | | |
| Illiterate | 51.6(205) | 52.3(149) | 50.0(56) | Ref | |
| Able to read and write | 17.6(70) | 17.2(49) | 18.7(21) | 0.89(0.48–1.59) | 0.666 |
| Primary education | 14.9(59) | 14.4(41) | 16.1(18) | 0.86(0.45–1.63) | 0.631 |
| Secondary education | 9.8(39) | 9.5(27) | 10.7(12) | 0.85(0.40–1.78) | 0.660 |
| Post-secondary education | 6.0(24) | 6.6(19) | 4.5(5) | 1.42(0.51–4.01) | 0.498 |
| **Household size (persons)** | | | | | |
| 1–3 | 91.9(365) | 93.7(267) | 87.4(98) | 3.81(1.18–7.30) | 0.025* |
| 4–6 | 5.0(20) | 4.6(13) | 6.3(7) | 2.60(0.60–6.31) | 0.203* |
| >6 | 3.0(12) | 1.7(5) | 6.3(7) | Ref | |
| **Presence of primary or secondary school student in household** | | | | | |
| Yes | 75.1(298) | 80.0(228) | 62.5(70) | 2.40(1.49–3.88) | <0.001* |
| No | 24.9(99) | 20.0(57) | 37.5(42) | Ref | |
| **Presence of children under five in household** | | | | | |
| No | 50.9(202) | 62.1(177) | 766.1(74) | 1.05(0.68–1.63) | 0.826 |
| Yes | 49.1(195) | 37.9(108) | 33.9(38) | Ref | |
| **Wealth index** | | | | | |
| Lowest | 26.2(104) | 24.2(69) | 31.2(35) | Ref | |
| Second | 29.0(115) | 29.8(85) | 26.8(30) | 1.44(0.80–2.57) | 0.222* |
| Middle | 20.2(80) | 20.7(59) | 18.8(21) | 1.43(0.74–2.71) | 0.280 |
| Fourth | 13.9(55) | 14.4(41) | 12.5(14) | 1.49(0.71–3.08) | 0.288 |
| Highest | 10.8(43) | 10.9(31) | 10.7(12) | 1.31(0.60–2.86) | 0.497 |

Ref, Reference category

*variables from bivariate analysis of p-value < 0.25 considered for multivariable analysis

**Table 2. Latrine characteristics and bivariate analysis with latrine utilization in semi-urban areas in northeastern Ethiopia, February to March, 2019.**

| Variables | | Latrine utilization (N = 397) | | COR (95% CI) | *p*-value |
|---|---|---|---|---|---|
| | Frequency | Utilized (*n* = 285) | Not utilized (*n* = 112) | | |
| | %(*n*) | %(*n*) | %(*n*) | | |
| **Type of latrine** | | | | | |
| Pit latrine with slab | 21.7(86) | 23.2(66) | 17.9(20) | 1.39(0.80–2.42) | 0.250* |
| Pit latrine without slab | 78.3(311) | 76.8(219) | 82.1(92) | Ref | |
| **Means of disposal of faeces of children under five** | | | | | |
| Pit latrine disposal | 145.3(180) | 45.6(130) | 44.6(50) | 0.92(0.57–1.50) | 0.714 |
| Disposal in the compound | 38.3(152) | 37.5(107) | 40.2(45) | 1.09(0.57–2.06) | 0.801 |
| Disposal outside the compound | 16.4(65) | 16.9(48) | 15.2(17) | Ref | |
| **Condition of latrine** | | | | | |
| Poor | 49.6(197) | 48.8(139) | 51.8(58) | Ref | |
| Fair | 38.3(152) | 37.9(108) | 39.3(44) | 1.02(0.64–1.63) | 0.920 |
| Good | 12.1(48) | 13.3(38) | 8.9(10) | 1.59(0.74–3.40) | 0.235* |
| **Faeces seen around pit hole/floor of latrine** | | | | | |
| No | 55.7(221) | 56.8(162) | 52.7(59) | 1.18 (0.76–1.18) | 0.453 |
| Yes | 44.3(176) | 43.2(123) | 47.3(53) | Ref | |
| **Distance of latrine from the house (meters)** | | | | | |
| < 6 | 3.5(14) | 3.1(9) | 4.5(5) | 0.57(0.17–1.98) | 0.380 |
| 6–50 | 80.9(321) | 80.4(229) | 82.1(92) | 0.79(0.42–1.49) | 0.474 |
| > 50 | 15.6(62) | 16.5(47) | 13.4(15) | Ref | |
| **Latrine location** | | | | | |
| Inside compound | 31.2(124) | 29.5(84) | 35.7(40) | 0.96(0.52–1.75) | 0.880 |
| Outside compound | 48.6(193) | 51.2(146) | 42.0(47) | 1.41(0.79–2.51) | 0.240* |
| No compound/latrine distant from home | 20.2(80) | 19.3(55) | 22.3(25) | Ref | |
| **Number of years since latrine was constructed** | | | | | |
| ≥ 2 | 43.8(174) | 47.7(136) | 33.9(38) | 1.78(1.23–2.80) | 0.013* |
| < 2 | 56.2(223) | 52.3(149) | 66.1(74) | Ref | |
| **Number of times latrine has been constructed** | | | | | |
| First time | 23.4(93) | 22.4(64) | 25.9(29) | Ref | |
| Second time | 47.1(187) | 46.7(133) | 48.2(54) | 1.12(0.65–1.92) | 0.691 |
| Third time | 16.9(67) | 18.6(53) | 12.5(14) | 1.72(0.82–3.56) | 0.150* |
| Fourth or more time | 12.6(50) | 12.3(35) | 13.4(15) | 1.06(0.50–2.23) | 0.884 |
| **Latrine squat hole covered** | | | | | |
| Yes | 37.0(147) | 37.5(107) | 35.7(40) | 1.08(0.69–1.70) | 0.734 |
| No | 63.0(250) | 62.5(178) | 25.3(72) | Ref | |
| **Latrine slab sealed with mud/cement** | | | | | |
| Yes | 21.7(86) | 25.8(62) | 21.4(24) | 1.02(0.56–1.72) | 0.943 |
| No | 78.3(311) | 78.2(223) | 78.6(88) | Ref | |
| **Latrine has walls** | | | | | |
| Yes | 75.1(298) | 80.0(228) | 62.5(70) | 2.4(1.49–3.88) | 0.001* |
| No | 24.9(99) | 20.0(57) | 37.5(42) | Ref | |
| **Latrine has a roof** | | | | | |
| Yes | 39.5(157) | 40.3(115) | 37.5(42) | 1.12 (0.72–1.77) | 0.601 |
| No | 60.5(240) | 59.6(170) | 62.5(70) | Ref | |
| **Latrine has a door** | | | | | |
| Yes | 74.1(294) | 78.2(223) | 63.4(71) | 2.08(1.29–3.35) | 0.003* |
| No | 25.9(103) | 21.8(62) | 36.6(41) | Ref | |

Ref, Reference category

*variables from bivariate analysis of *p*-value < 0.25 considered for multivariable analysis

who had constructed a latrine for the third time were 1.72 times (COR: 1.72, 95% CI [0.82–3.56]) more likely to utilize their latrine than those who had constructed a latrine for the first time (Table 2).

### Behavioral characteristics

Of the total study participants, 30.2% (120) of households reported cleaning the latrine rarely, whereas one-fourth (24.2%) of the households cleaned the latrine daily. The hygienic condition of a majority 82.9% (329) of the latrines was dirty and 17.1% (68) clean. Three-fourth (74.3%) of the study participants had received information about constructing latrines. The majority 62.5% (248) of households constructed latrines because of the advice of health extension workers and about one-fourth 23.4% (93) constructed latrines on their own initiative. About 45.8% (182) of the study participants considered that lacking a latrine was a cultural taboo; the types of the taboo were *shem* (46.7%), presence of bad smell (31.3%), fly problem (11.5%) and disease problem (10.4%) (Table 3).

From the bivariate analysis, households that cleaned their latrine daily were 1.93 times (COR: 1.93, 95% CI [1.02–3.67]) more likely to use it compared with those that cleaned their latrine rarely. The odds of latrine utilization among study participants who had received information about latrine construction were 2 times (COR: 2.0, 95% CI [1.24–3.22]) higher than among those who had not received information about latrine construction (Table 3).

### Prevalence of latrine utilization

The prevalence of latrine utilization was 71.8% (95% CI [67.5–76.1%]), whereas 28.2% (95% CI [23.9–32.5%]) of participants did not utilize latrines (Fig 1).

### Factors associated with latrine utilization

The multivariable analysis revealed that household family size of one to three persons, presence of primary or secondary school student in the house, number of years since construction of the latrine equal or greater than two and daily cleaning of latrine were significantly associated with latrine utilization. The odds of latrine utilization of households that had one to three family members were 3.99 times (AOR: 3.99, 95% CI [1.20–6.24]) higher than of households of greater than six members. The study also revealed that the odds of latrine utilization for households that included a primary or secondary school student were 2.33 times (AOR: 2.33, 95% CI [1.42–3.83]) higher than for those that did not include a primary or secondary school student.

Furthermore, the odds of latrine utilization in households in which it had been two or more years since the latrine had been constructed were 1.82 times (AOR: 1.82, 95% CI [1.12–2.95]) higher than for households in which it had been constructed more recently. The odds of latrine utilization for households that cleaned the latrine daily were 2.19 times (AOR: 2.19, 95% CI [1.12–4.28]) higher than for households that cleaned their latrine rarely (Table 4).

### Discussion

This community-based cross-sectional study found that nearly three-fourths of participants utilized latrines, and that latrine usage was significantly associated with a household family size of one to three persons, presence of primary or secondary school student in the household, time since the construction of the latrine of two or more years and daily cleaning of the latrine.

In this study, the rate of latrine utilization was lower than found in a study done in Wondo Genet in SNNPs (South Nation Nationalities and Peoples), Ethiopia [24] and Hotesa Arisi

**Table 3. Behavioral characteristics and bivariate analysis with latrine utilization in semi-urban areas in northeastern Ethiopia, February to March, 2019.**

| Variables | Latrine utilization (N = 397) | | | COR (95% CI) | p-value |
|---|---|---|---|---|---|
| | Frequency | Utilized (n = 285) | Not utilized (n = 112) | | |
| | %(n) | %(n) | %(n) | | |
| **Frequency of latrine cleaning** | | | | | |
| Weekly | 24.7(98) | 22.4(64) | 30.4(34) | 0.83(0.47–1.48) | 0.545 |
| Daily | 24.2(96) | 27.4(78) | 16.1(18) | 1.93(1.02–3.67) | 0.045* |
| When dirty | 20.9(83) | 21.1(60) | 20.5(23) | 1.16(0.62–2.15) | 0.632 |
| Rarely | 30.2(120) | 29.1(83) | 33.0(37) | Ref | |
| **Latrine hygienic condition** | | | | | |
| Clean | 17.1(68) | 48(16.8) | 17.9(20) | 0.93(0.52–1.67) | 0.809 |
| Dirty | 82.9(329) | 83.2(237) | 82.1(92) | Ref | |
| **Information received about constructing latrine** | | | | | |
| Yes | 74.3(295) | 78.2(223) | 64.3(72) | 2.0(1.24–3.22) | 0.005* |
| No | 25.7(102) | 21.8(62) | 35.7(40) | Ref | |
| **Reasons to construct latrine** | | | | | |
| Advice by health extension worker | 62.5(248) | 62.1(177) | 63.4(71) | 1.13(0.38–3.39) | 0.823 |
| From seeing others build | 10.1(40) | 9.1(26) | 12.5(14) | 0.84(0.24–2.92) | 0.789 |
| Self-initiated | 23.4(93) | 24.9(71) | 19.6(22) | 1.47(0.46–4.68) | 0.517 |
| Imposition from *kebele* | 4.0(16) | 3.9(11) | 4.5(5) | Ref | |
| **Person responsible for constructing latrine** | | | | | |
| Men | 33.5(133) | 33.7(96) | 33.0(37) | 1.04(0.63–1.7) | 0.884 |
| Women | 20.7(82) | 20.0(59) | 20.5(23) | 1.02(0.57–1.83) | 0.931 |
| Both | 45.8(182) | 45.6(130) | 46.5(52) | Ref | |
| **Lack of latrine considered culturally taboo** | | | | | |
| No | 54.2(215) | 54.4(155) | 53.6(60) | 1.03(0.67–1.60) | 0.883 |
| Yes | 45.8(182) | 45.6(130) | 46.4(52) | Ref | |
| **Type of taboo** | | | | | |
| *Shem* | 46.7(85) | 50.8(66) | 36.5(19) | 1.63(0.55–4.86) | 0.383 |
| Presence of bad smell | 31.3(57) | 28.5(37) | 38.5(20) | 0.85(0.28–2.59) | 0.780 |
| Fly problem | 11.5(21) | 10.8(14) | 13.5(7) | 0.92(0.25–3.48) | 0.906 |
| Disease problem | 10.4(19) | 10.0(13) | 11.5(6) | Ref | |
| **Presence of handwashing facility near latrines** | | | | | |
| No | 138.8(154) | 38.9(111) | 38.4(43) | 1.02(0.65–1.60) | 0.919 |
| Yes | 61.2(243) | 61.1(174) | 61.6(69) | Ref | |

Ref, Reference category

*variables from bivariate analysis of p-value < 0.25 considered for multivariable analysis

District, Oromia Region in Ethiopia [25], but higher than found in various other rural areas of Ethiopia such as Anded [26], Denbia [11], Chencha [12], Enderta [27], Laelai Maichew District [28] and by national systematic review and meta-analysis survey of latrine utilization [29]. The relatively higher prevalence of latrine utilization in our study compared with the rural areas mentioned might be due to residents of this semi-urban area having a better awareness of latrine utilization, sanitation and hygiene practices, education opportunities and the presence of government employees in Alansha having a positive influence on latrine utilization.

Our study indicated that a household family size of one to three persons was one of the determinant factor for latrine utilization, a finding consistent with similar studies in other areas of Ethiopia such as in Southeast Zone of Tigray [30] and in Hawassa [31]. Sharing one

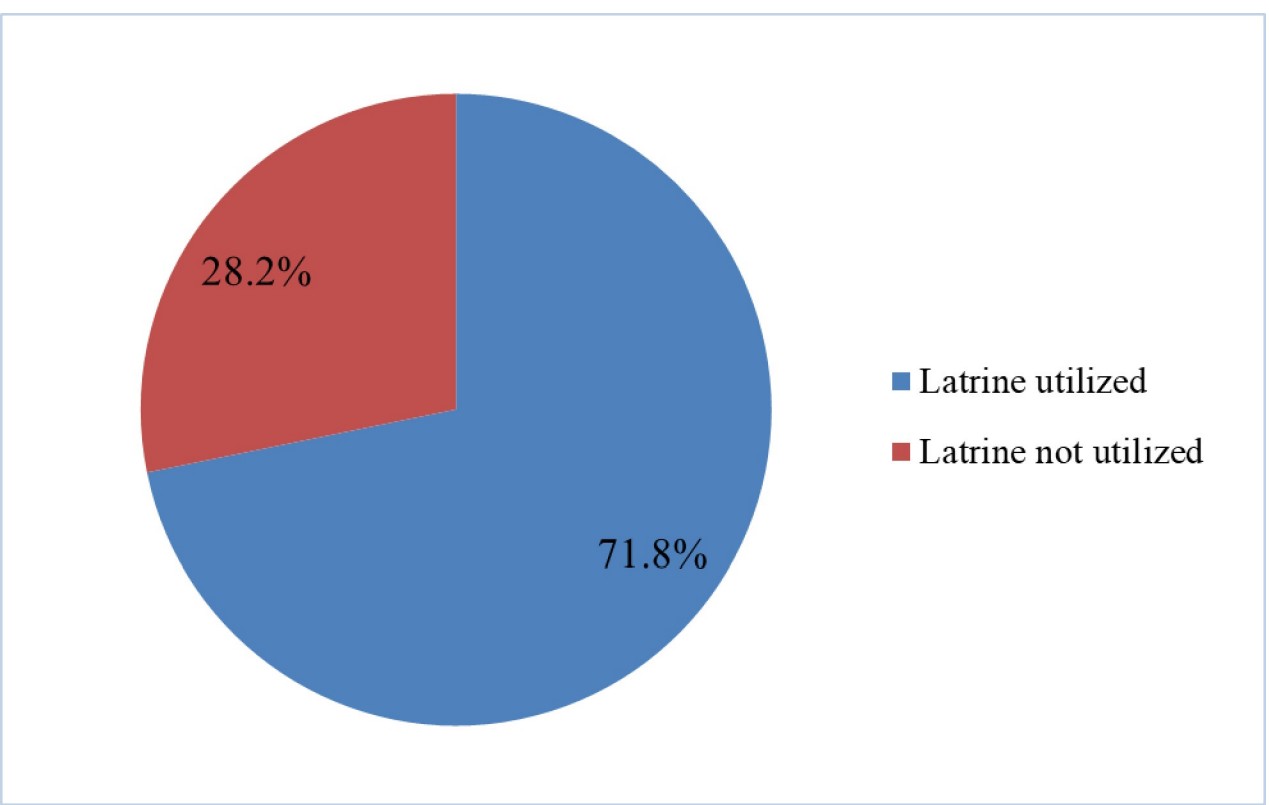

**Fig 1. Latrine utilization status in semi-urban areas of Alansha, South Wollo Zone, northeastern Ethiopia, February to March, 2019.**

latrine among fewer family members results in the latrine being used less frequently overall, making the latrine more likely to be cleaner, which in turn may increase latrine utilization. However, sharing of a latrine by a large family increases the number of times the latrine is used on a daily basis, thereby making the latrine more likely to be dirty, which in turn may decrease utilization of the latrine. A study in slums of Addis Ababa showed that a large family size of six or more persons led to poor hygiene of the latrines, which was in turn associated with diarrhea among under-five children [20] and that the presence of continuously available latrines helped to control the diarrheal disease [32]. The presence of a larger family size may compromise an individual's feeling of responsibility to use the latrine properly, another possible reason for latrines to be dirty, as this study revealed.

Our study also revealed that the presence of a primary or secondary school student in a household increased the odds of latrine utilization, which is consistent with similar studies in other areas of Ethiopia such as in Hullet Eju Enessie in Gojjam [19], Laelai Maichew District in Tigray [30] and Denbia district in Gondar [11]. This might be due to the fact that primary or secondary school students were more exposed to hygiene information in the school environment and therefore their presence positively favored latrine utilization at home. A study in Uganda also found that the presence of a primary or secondary school student positively favored latrine utilization in the home environment [33]. The study area district administrator reported (via personal communication) that the health extension program was closely linked to the promotion of health at school, which was an additional opportunity for students to learn healthy lifestyles. Another study revealed that the active involvement of health professionals in latrine hygiene and sanitation is crucial to accelerating and consolidating progress towards the desired goals [34].

**Table 4. Factors significantly associated with latrine utilization from multivariable logistic regression analysis in semi-urban areas in Northeastern Ethiopia, February to March, 2019.**

| Variables* | AOR (95% CI) |
|---|---|
| **Household size (persons)** | |
| 1–3 | **3.99(1.20–6.24)** |
| 4–6 | 3.05(0.67–5.90) |
| >6 | Ref |
| **Presence of primary or secondary school student in a household** | |
| Yes | **2.33(1.42–3.83)** |
| No | Ref |
| **Number of years since latrine constructed** | |
| ≥ 2 | **1.82(1.12–2.95)** |
| < 2 | Ref |
| **Cleaning frequency of latrine** | |
| Weekly | 1.03(0.57–1.89) |
| Daily | **2.19(1.12–4.28)** |
| When dirty | 1.33(0.69–2.54) |
| Other (rarely) | Ref |

Ref, Reference category; AOR, Adjusted odds ratio; CI, Confidence interval.

*Variables adjusted for multivariable analysis were study participant age, occupation, household size, presence of primary or secondary school student in the household, wealth index, type of latrine, condition of latrine, latrine location, years since latrine constructed, number of times

Latrine had been constructed, latrine having wall, door, frequency of latrine cleaning, information received about constructing latrine and presence of water in handwashing facility.

Our study showed that households in which it had been two and more years since the latrine was constructed were more likely to utilize their latrine than those in which the latrine had been constructed more recently. Findings consistent with ours are mentioned in studies of Ethiopian areas including Aneded district [26], Dilla town [35], Chencha [12] and in Wondo Genet [24]. The number of years since the construction of a latrine being a factor for latrine utilization might be due to the fact that attainment of behavioral change among household members may take some period of time. Therefore, to improve latrine utilization, a continuous effort in education, support and monitoring of latrines needs to be regularly maintained until community behavioral change is observed in a sustained manner. Similar with our study findings, a study conducted in India also showed that more years since latrine construction increased the utilization of the latrine [36].

This study also revealed that daily cleaning of the latrine was significantly associated with latrine utilization, a finding similar to several Ethiopian studies in Wondo Genet [24], Hetosa district [25], Laelai Maichew [30] and Aneded district [26]. In addition, our finding was consistent with a study in Kenya [37]. When latrines are cleaned frequently, faeces, flies and bad odors are eliminated, all of which may increase latrine utilization.

## Limitations of the study

One of the limitations of this study was that data was obtained from a cross-sectional survey study, which may be exposed to bias due to self-reporting. During self-reporting, there may be an occurrence of social desirability bias [38]. In the absence of follow-up observational data, this work may greatly underestimate or overestimate the magnitude of latrine utilization and other independent variables. Although the latrine utilization during the study period was

determined by using on-the-spot- observation, it was difficult to determine whether there was consistent use of the latrine using a cross-sectional study.

Another limitation of this cross-sectional study was the difficulty of establishing causal relationships between the latrine utilization status and independent factors. Also, our study was conducted during February and March, a period that is in a relatively dry season in the study area, and further studies that considered latrine utilization during seasonal variation is recommended. Furthermore, the results of this study may not be representative of the occurrence and underlying factors of latrine utilization across all semi-urban areas in northeastern Ethiopia due to the study being conducted only in small semi-urban areas.

## Conclusion

Based on the findings, we concluded that a majority of households utilized a latrine. From this study, we concluded that factors significantly associated with latrine utilization were a household family size from one to three persons, presence of primary or secondary school student in the household, number of years since latrine construction (two years or more) and daily cleaning of the latrine. It is recommended that promotion of behavioral change toward sustainable and consistent latrine use should be carried out as an essential step for further improvement of latrine utilization status. Further observational research triangulated with qualitative study should be conducted to provide more strong evidence for further improvement of household latrine utilization status.

## Supporting information

**S1 Data.**
(XLSX)

**S1 Questionnaire.**
(DOCX)

**S2 Questionnaire.**
(DOCX)

## Acknowledgments

We acknowledge Kutaber District, Alansha *kebele* and South Wollo Zone municipality administrators for their kind assistance in providing necessary information and support during data collection. We also thank the study participants, data collectors, and supervisors for their dedication and commitment during the study. We also thank Lisa Penttila for language editing of the manuscript.

## Author Contributions

**Conceptualization:** Deres Asnake, Metadel Adane.

**Data curation:** Deres Asnake, Metadel Adane.

**Formal analysis:** Deres Asnake, Metadel Adane.

**Funding acquisition:** Deres Asnake, Metadel Adane.

**Investigation:** Deres Asnake, Metadel Adane.

**Methodology:** Deres Asnake, Metadel Adane.

**Project administration:** Deres Asnake, Metadel Adane.

**Resources:** Deres Asnake, Metadel Adane.

**Software:** Deres Asnake, Metadel Adane.

**Supervision:** Metadel Adane.

**Validation:** Metadel Adane.

**Visualization:** Metadel Adane.

**Writing – original draft:** Metadel Adane.

**Writing – review & editing:** Metadel Adane.

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
