## [Decision Letter · Decision Letter 0]

6 May 2020

PONE-D-20-01859

Household Latrine Utilization and Its Association with Household Family Size in Semi-Urban Areas of Alansha, South Wollo, Northeastern, Ethiopia

PLOS ONE

Dear Dr Adane (PhD),

Thank you for submitting your manuscript to PLOS ONE. After careful consideration, we feel that it has merit but does not fully meet PLOS ONE’s publication criteria as it currently stands. Therefore, we invite you to submit a revised version of the manuscript that addresses the points raised during the review process.

Your manuscript has been assessed by two reviewers, who request a number of revisions to clarify the number of households included in the analysis and to place the findings in the appropriate context (notably, ensuring that relevant information from the 2016 EDHS is discussed). In addition, please provide a copy of the questionnaire as supplementary information, and that information about how/whether the questionnaire was validated is provided in the revised manuscript. Finally, please ensure that the manuscript is copyedited prior to resubmission to address concerns about clarity and precision as noted by the reviewers, and to resolve the instances of text overlap that were identified by the staff editors.

We would appreciate receiving your revised manuscript by Jun 19 2020 11:59PM. To enhance the reproducibility of your results, we recommend that if applicable you deposit your laboratory protocols in protocols.io, where a protocol can be assigned its own identifier (DOI) such that it can be cited independently in the future. For instructions see: http://journals.plos.org/plosone/s/submission-guidelines#loc-laboratory-protocols

We look forward to receiving your revised manuscript.

Kind regards,

Emily Chenette

Staff Editor

PLOS ONE

Journal Requirements:

2.  We noticed you have some minor occurrence(s) of overlapping text with the following previous publication(s), which needs to be addressed:

https://www.ajol.info/index.php/ejhd/article/view/62959

https://doi.org/10.12691/ajphr-5-4-2

https://www.dx.doi.org/10.11604/pamj.26/08/2014.18.334.4206

https://doi.org/10.4172/2167-1095.1000174

https://doi.org/10.5897/AJEST2016.2223

https://dhsprogram.com/pubs/pdf/FR328/FR328.pdf

https://doi.org/10.1186/s13104-019-4684-3

In your revision ensure you cite all your sources (including your own works), and quote or rephrase any duplicated text outside the Methods section. Further consideration is dependent on these concerns being addressed.""

3. Please include additional information regarding the survey or questionnaire used in the study and ensure that you have provided sufficient details that others could replicate the analyses. For instance, if you developed a questionnaire as part of this study and it is not under a copyright more restrictive than CC-BY, please include a copy, in both the original language and English, as Supporting Information.""

4. Please provide additional details regarding participant consent. In the ethics statement in the Methods and online submission information, please ensure that you have specified whether consent was written or verbal/oral. If consent was verbal/oral, please specify: 1) whether the ethics committee approved the verbal/oral consent procedure, 2) why written consent could not be obtained, and 3) how verbal/oral consent was recorded. If your study included minors, please state whether you obtained consent from parents or guardians in these cases. If the need for consent was waived by the ethics committee, please include this information.

'We appreciate the support of Amhara Regional Government Health Bureau by providing for funds for this study.'

'The funders had no role in study design, data collection and analysis, decision to publish, or preparation of the manuscript.'

Additional Editor Comments (if provided):

Reviewers' comments:

Reviewer's Responses to Questions

**Comments to the Author**

1. Is the manuscript technically sound, and do the data support the conclusions?

Reviewer #1: Yes

Reviewer #2: Yes

2. Has the statistical analysis been performed appropriately and rigorously? 

Reviewer #1: No

Reviewer #2: Yes

3. Have the authors made all data underlying the findings in their manuscript fully available?

Reviewer #1: Yes

Reviewer #2: Yes

4. Is the manuscript presented in an intelligible fashion and written in standard English?

Reviewer #1: Yes

Reviewer #2: Yes

5. Review Comments to the Author

Reviewer #1: This manuscript accepted with major comments.

Overall this is a good study but needs to be reviewed and toned down by the authors. The author need to focus and re-write the result section and should present the results more clearly and in a standard way.

Reviewer #2: Review report (Reviewer # 1 Habtamu Tolera)

The manuscript reports the findings regarding Household Latrine utilization and its association with household family size in semi-urban areas of Alansha, south Wollo, northeastern, Ethiopia. This kind of study is much relevant in context of developing countries like Ethiopia. The study has large sample and tried to represent each unit from the study setting Analysis has been made well and the authors have made relevant conclusions/recommendations based on the findings. However, I have some concerns which should be addressed and revised accordingly to get published in PLoS ONE. You can also refer to the attached PDF file.

Abstract

Page # 2 line 30, under the methods section it was said that “a cross-sectional study was conducted … among 397 households”. Likewise, on page # 2 line 40 under the same section author (s) also mentioned a total of 401 households … were assessed ...” Why different figures? This needed to be corrected. On page # 2 line 38 you should delete the unnecessary phrase used in the bracket which is called “significant at” because Р-value is enough to describe the significance of the model high level scholars or readers.

Background

On page # 2 lines 64 and 65 needs you citing source (s). This concern works for all if such cases were there elsewhere across the text.

On page 4 line 66 and 67 you described that “In 2010, 67 15% of the population still practiced open defecation” Why did not refer recent global report or figure?

PLoS ONE follows Vancouver style for in text-citation, should be indicated by the reference number in rectangular brackets but authors used APA style across the entire manuscript. Suiting journal’s instructions is vital.

On page # 4 line 69 you should have to discuss Ethiopian (local) context regarding the prevalence of open defection. Use at least 2016 EDHS report, the proportion of overall population in the country who use open defecation or open field than modern latrine facilities. On page 4 lines 74 and 75 you described that “Lack of latrine utilization has been due mainly to, …” You need to acknowledge or cite original source(s).

On page # 4 lines 80-82 seems concluding point. Put it under “conclusion section” in the last page of the manuscript.

Page # 5 Line 95 delete the phrase “recent data from the”.

On page # 5 Line 100 you need to mention EDHS 2016 as it was the most recent survey than the Mini DHS of 2014. so better to describe like this, “The recent data from the Ethiopia Demographic and Health Survey (EDHS) (2016) indicated that 6% of Ethiopian households…”

Materials and methods

On page # 5 line 116, under a section called study setting “which is located….” is appropriate.

Delete the outcome variable from the sub-section “Study design and outcome variable”. Move your whole discussion about outcome variable and insert before a sub-section “Independent variable”, separately or you can merge both outcome and independent variables into one under a sub-tittle “Variables measured”. Hence, move the whole texts from lines 126 through 134 there.

Page # 6 lines 129 and 130 you mentioned, “…World Health Organization (WHO) and Joint Monitoring Program (JMP) “signs of use …” . It needs you citing the original source

Consistency should be kept across the manuscript, e.g. on page 6 line 116 you used woreda but on page # 4 line 85 I have seen “In Denbia District” and elsewhere you also used the word “district”, I advise you use either of the two, “Woreda “(should be italicized) or “District” .

Page # 9 line 178, you mentioned that,” …Environmental Health” use lower case. Page #9, line 180, rewrite the sentence like this “The principal investigator provided a training that took the whole one day…”

Page # 9 lines 191 through 193 does not convey a clear message for readers. Please rewrite this sentence

On page # 10 under a section Independent variable you missed to list the Households wealth status distributions across five categories: poorest, second, middle, fourth and highest should be described as an exposure variable. you need to discuss the issue here as well

On page # 10 line 208, a kind of ambiguity happens there, so delete “other” and use instead “Ash” or use “other” and define what it features in bracket

Ethical consideration needs citing of ECR’s protocol number of the approval letter.

Move the sentence on page # 11 from lines 216 through 218 to the last section of this paragraph.

Results

Rewrite line 239 like this, “In this study, of the total 401 participants,”, Line 240 like this, “Out of this, the majority (71.8%),...”. Line 241 like this, “Across gender, of the total 397 respondents,”. Line 243 lie this, “who were Muslims were…”, “other religion followers were…”

Table caption for Table 1 should be placed in the text immediately after you referenced the table in the in-text citation, in the last paragraph of your respective discussion about the table. This comment works for the remaining tables in the entire manuscript or elsewhere in the text.

On page # 12, rewrite the sentence from lines 247 through 250. Correct line 252 like this, “Of the total respondents,30.2% of households reportedly showed that they had a cleaning frequency (of latrines) rarely while 24.7% reported that they made every day”. Please line 253 is not clear for me rewrite.

Discussion

On page 14 line 282, rewrite like this, “ in this study, the rate of latrine facility utilization…”. Line 283 and 284 like this, “elsewhere in other parts of Ethiopia such as...”. Line 285 through 290 is too long sentence. Please keep it succinct and to the point.

Try to rewrite a sentence from line 292 through 295 like this, After adjusting for all variables, a socio-demographic factors in a form of households with small family size of 1-3 members were significantly associated with latrine facility utilization as opposed to households with larger family members (4 person or more), which is consistent with studies in southeastern zone of Tigray(), and in Hawassa ( )

Lines 297 through 299 is not clear, rewrite this paragraph.

Conclusion

In conclusion, there is no need to make discussion, just write the major conclusions and what recommendations they could made for policy makers. Your conclusion under abstract section is better stated than the one that was found at the last page of the manuscript. Move the concluding points under Abstract section to your Major “Conclusion” section in the last page of the manuscript.

Acknowledgement

In PLoS ONE, you acknowledge funders not under Acknowledgment section but under funding section during online submission.

Reference list

Change reference lists into Vancouver style please. Avoid capitalization problems as well. Do according to the Journal’s guideline

Overall, some editorial works are here in the manuscript. It should be made more concise and clearer in each section. They should check and revised thoroughly.

6. PLOS authors have the option to publish the peer review history of their article (what does this mean?). If published, this will include your full peer review and any attached files.

Reviewer #1: Yes: Mahfuza Islam

Reviewer #2: Yes: Habtamu Tolera

---

## [Author Response · Author response to Decision Letter 0]

30 Jul 2020

Date: 31 July, 2020

Manuscript ID: PONE-D-20-01859R1

Household latrine utilization and associated factors in Semi-Urban areas in Northeastern Ethiopia

Corresponding authors: Metadel Adane (PhD)

Dear Emily Chenette, staff editor, PLoS ONE 

Thank you for your letter dated May 06, 2020 with a decision of revision needed. We were pleased to know that our manuscript was considered potentially acceptable for publication in PLoS ONE, subject to adequate revision as requested by the reviewers. Based on the instructions provided in your letter, we uploaded the file of the rebuttal letter; the marked up copy of the revised manuscript highlighting the changes made in the original submitted version and the clean copy of the revised manuscript. 

We have revised the manuscript by modifying the abstract, introduction, methods, results, discussion and other sections, based on the comments made by the reviewers and using the journal guidelines. Accordingly, we have marked in red color all the changes made during the revision process. Appended to this letter is our point-by-point response (rebuttal letter) to the comments made by the reviewers. 

We agree with almost all the comments/questions raised by the reviewers and provided justification for disagreeing with some of them. We would like to take this opportunity to express our thanks to the reviewers for their valuable comments and to thank you for allowing us to resubmit a revision of the manuscript. 

Funding

Amhara Region Health Bureau funded this study. The funders had no role in study design, data collection and analysis, decision to publish, or preparation of the manuscript. 

I hope that the revised manuscript is accepted for publication in PLoS ONE. 

Rebuttal letter

Response to the Journal Requirements 

Question #1: Please ensure that your manuscript meets PLOS ONE's style requirements, including those for file naming

Response: Thank you for this remark. We re-formatted the revised manuscript using the PLoS ONE format guidelines. The whole content of the manuscript, including the abstract, introduction, methods, discussion and reference are formatted using the guidelines (please see the revised version for each section).

Comment #2. We noticed you have some minor occurrence (s) of overlapping text with the following previous publication (s), which needs to be addressed: In your revision ensure you cite all your sources (including your own works), and quote or rephrase any duplicated text outside the Methods section. Further consideration is dependent on these concerns being addressed. 

Response: We addressed this issue seriously and please check the revised version. 

Comment #3. Please include additional information regarding the survey or questionnaire used in the study and ensure that you have provided sufficient details that others could replicate the analyses. For instance, if you developed a questionnaire as part of this study and it is not under a copyright more restrictive than CC-BY, please include a copy, in both the original language and English, as Supporting Information.

Response: We provided the questionnaire in original language and English version as supporting information S1 and S2 (please see the revised version attachment). 

Comment #4. Thank you for stating the following in the Acknowledgments Section of your manuscript: We appreciate the support of Amhara Regional Government Health Bureau by providing for funds for this study.' We note that you have provided funding information that is not currently declared in your Funding Statement. However, funding information should not appear in the Acknowledgments section or other areas of your manuscript. We will only publish funding information present in the Funding Statement section of the online submission form. Please remove any funding-related text from the manuscript and let us know how you would like to update your Funding Statement. Currently, your Funding Statement reads as follows: 'The funders had no role in study design, data collection and analysis, decision to publish, or preparation of the manuscript.'

Response: Thank you for these pertinent comments. We made the necessary changes and we noted the funding statement in the cover letter. 

Comment #5. - Please provide additional details regarding participant consent. In the ethics statement in the Methods and online submission information, please ensure that you have specified how verbal consent was documented and witnessed.

Response: We provided all the ethical consideration issues of the oral/verbal consent statement and please see the revised version (see the revised version). 

Line by line response to reviewers

Reviewer # 1

Comments on Manuscript:

Overall this is a good study but needs to be reviewed and toned down by the authors for the following checks:

Response: Thank you for the positive remark on our manuscript and we made all the corrections throughout the manuscript as follow. 

Title:

The authors may revise the language of the title/s to improve readability. Author may consider writing the title as “Household Latrine Utilization and Its Association with Household Family Size in Semi-Urban Areas in Ethiopia”. 

Response: We revised the title almost similar with the reviewer comment. The modified title is “Household latrine utilization and associated factors in Semi-Urban areas in Northeastern Ethiopia”. This title is directly fits the content of the paper. 

Abstract:

1) Author reported a different number of households in method and result of the abstract. In line 30, number of households mentioned 397 but in line 40, it was mentioned 401. Later on, in line 151 final sample sizes mentioned as 401. I would suggest checking the number and making it clear.

Response: Thank you for the comment and the error is corrected (please see the revised version). 

2. In line 43, need a space between 285 and (71.8%).

Response: We add space as suggested. 

3. In line 42 put a coma (,) instead of full stop (.) after the value 3.99

Response: We put comma and sorry for this error. 

4. In line 43, need a space between constructed and ≥2 years. Author made the same error in line 240-242 as well as in many lines of the manuscript and in the table of result. I would suggest checking for the same error throughout the manuscript. 

Response: Thank you and we updated and checked similar errors throughout the paper. 

Materials and Methods

5. In line 176, author may consider to write “spot check observation” instead of “observation”.

Response: Thank you for this pertinent comment. We revised as suggested (See the revised version throughout the paper in lines 185). 

6. In line 207, what do author mean by “kebele”? Make it clear.

Response: We defined kebele in the study area sub-heading to make it clear for readers. kebele is the smallest administrative unit in Ethiopia consisted of an average 5,000 populations (See the revised version in lines 109 to 111). 

Results

I would recommend the author to re-write the result section including the results Tables (Table1 to table 4). Author need to work on the following points:

Response: We updated Tables 1 to 4 (See all the revised tables). 

7. Report the prevalence and keep the number within bracket, eg 71.8% (285). Make the changes throughout the result section. In addition, remove the decimal if the prevalence more than 10%. Author should report the bivariate result in the result section.

Response: We thank for this pertinent comment and we revised as suggested. However we reported the decimal as it is without approximation since to be consistent with throughout the paper. 

8. In line 247, re-write the result to improve readability. 

Response: The sentence is revised to make it clear. 

9. In line 261, check the value of 95% CI. A proof reading is must for the author, especially in the result section.

Response: Thank you. We corrected the errors. 

10. The title of the tables is too long. I would recommend the author to make it short. Better to remove the location and duration from the title, eg. “Table 1: Socio-demographic characteristics and bivariate analyses with latrine utilization in semi-urban area in Ethiopia”.

Response: The title of Table 1 to Table 4 is revised to make it short and precise (See the titles of Table 1 to Table 4). 

11. Author may consider bold the significant results in the tables.

Response: For Table 4, we make it bold for the significant results (See Table 4). However, for other Tables, being significant from the bivariate analysis is not important due to the confounders not controlled (See Table 4). 

12. In line 423-424 (Table 1), I would recommend the author to report prevalence for the column utilized and not utilized. Same recommendation for the rest of the tables. Without prevalence it’s difficult to check the bivariate results.

Response: We agree with the given comments and we reported the prevalence for column as suggested (See Table 1). 

13. In line 454-455 (Table 3), author mentioned “n (%)” in 3rd and 4th column but only report the frequency. I would recommend to report both frequency and %.

Response: Thank you for this key comment. We revised all tables and the frequency and percentage was estimated (See Table 3). 

Discussion

14. In line 278, Author should remove number (prevalence or 95% CI) form the discussion section. It’s not a good practice to report the result in the discussion. 

Response: Thank you, we updated as suggested. 

15. Author should work on the discussion depending on the changes of result section.

Response: We revised the discussion as suggested (please see the revised version of the discussion in pages 321 to 397). 

Reviewer #2

 Review report (Reviewer # 2 Habtamu Tolera)

The manuscript reports the findings regarding Household Latrine utilization and its association with household family size in semi-urban areas of Alansha, south Wollo, northeastern, Ethiopia. This kind of study is much relevant in context of developing countries like Ethiopia. The study has large sample and tried to represent each unit from the study setting Analysis has been made well and the authors have made relevant conclusions/recommendations based on the findings. However, I have some concerns which should be addressed and revised accordingly to get published in PLoS ONE.

Abstract

Page # 2 line 30, under the methods section it was said that “a cross-sectional study was conducted … among 397 households”. Likewise, on page # 2 line 40 under the same section author (s) also mentioned a total of 401 households … were assessed ...” Why different figures? This needed to be corrected. On page # 2 line 38 you should delete the unnecessary phrase used in the bracket which is called “significant at” because Р-value is enough to describe the significance of the model high level scholars or readers. 

Response: This error is corrected (See abstract). 

Background

On page # 2 lines 64 and 65 needs you citing source (s). This concern works for all if such cases were there elsewhere across the text.

Response: Thank you and we cited. 

On page 4 line 66 and 67 you described that “In 2010, 67 15% of the population still practiced open defecation” Why did not refer recent global report or figure?

Response: It is deleted. 

PLoS ONE follows Vancouver style for in text-citation, should be indicated by the reference number in rectangular brackets but authors used APA style across the entire manuscript. Suiting journal’s instructions is vital.

Response: 

On page # 4 line 69 you should have to discuss Ethiopian (local) context regarding the prevalence of open defection. Use at least 2016 EDHS report, the proportion of overall population in the country who use open defecation or open field than modern latrine facilities. 

Response: It is revised as recommended focusing to Ethiopian Open defecation status (see lines 73 to 75). 

On page 4 lines 74 and 75 you described that “Lack of latrine utilization has been due mainly to, …” You need to acknowledge or cite original source(s). 

Response: 

On page # 4 lines 80-82 seems concluding point. Put it under “conclusion section” in the last page of the manuscript. 

Response: It is moved to the conclusion section (see the conclusion). 

Page # 5 Line 95 delete the phrase “recent data from the”. 

Response: Deleted 

On page # 5 Line 100 you need to mention EDHS 2016 as it was the most recent survey than the Mini DHS of 2014. so better to describe like this, “The recent data from the Ethiopia Demographic and Health Survey (EDHS) (2016) indicated that 6% of Ethiopian households…”

Response: Thank you, it is updated. 

 Materials and methods

On page # 5 line 116, under a section called study setting “which is located….” is appropriate.

Response: Thank you, it is changed. 

Delete the outcome variable from the sub-section “Study design and outcome variable”. Response: Yes, it is deleted 

Move your whole discussion about outcome variable and insert before a sub-section “Independent variable”, separately or you can merge both outcome and independent variables into one under a sub-tittle “Variables measured”. Hence, move the whole texts from lines 126 through 134 there. 

Response: It sub-titled as variables measured (see line 158). 

Page # 6 lines 129 and 130 you mentioned, “…World Health Organization (WHO) and Joint Monitoring Program (JMP) “signs of use …” . It needs you citing the original source

Response: It is deleted since it is not the correct source. 

Consistency should be kept across the manuscript, e.g. on page 6 line 116 you used woreda but on page # 4 line 85 I have seen “In Denbia District” and elsewhere you also used the word “district”, I advise you use either of the two, “Woreda “(should be italicized) or “District”. 

Response: It changed to district throughout the paper (see the revised version). 

Page # 9 line 178, you mentioned that,” …Environmental Health” use lower case. Page #9, line 180, rewrite the sentence like this “The principal investigator provided a training that took the whole one day…”

Response: Yes, Environmental Health changed to lower case letters as environmental health (See line 194). 

Page # 9 lines 191 through 193 does not convey a clear message for readers. Please rewrite this sentence.

Response: It is re-written, please see the revised version 

On page # 10 under a section Independent variable you missed to list the Households wealth status distributions across five categories: poorest, second, middle, fourth and highest should be described as an exposure variable. You need to discuss the issue here as well.

Response: We put the wealth categories of lowest, second, middle, fourth and highest (see the revised version in lines 176 to 177). 

On page # 10 line 208, a kind of ambiguity happens there, so delete “other” and use instead “Ash” or use “other” and define what it features in bracket.

Response: we deleted all to avoid confusions. 

Ethical consideration needs citing of ECR’s protocol number of the approval letter.

Response: We cited the protocol number (See line 231). 

Move the sentence on page # 11 from lines 216 through 218 to the last section of this paragraph.

Response: We provided the details for verbal/oral consent. Please see the revised version at the ethical consideration section (See lines 230 to 240). 

Results

Rewrite line 239 like this, “In this study, of the total 401 participants,”, Line 240 like this, “Out of this, the majority (71.8%),...”. Line 241 like this, “Across gender, of the total 397 respondents,”. Line 243 lie this, “who were Muslims were…”, “other religion followers were…”

Response: We revised the sentence to make it more clear (See lines 244 to 247). 

Table caption for Table 1 should be placed in the text immediately after you referenced the table in the in-text citation, in the last paragraph of your respective discussion about the table. This comment works for the remaining tables in the entire manuscript or elsewhere in the text.

Response: As per the format for PLoS ONE, caption inside the text is needed for figures but for Tables captions is needed in the title of the table. We did as per the guidelines. 

On page # 12, rewrite the sentence from lines 247 through 250. Correct line 252 like this, “Of the total respondents,30.2% of households reportedly showed that they had a cleaning frequency (of latrines) rarely while 24.7% reported that they made every day”. Please line 253 is not clear for me rewrite. 

Response: We did the amendment as suggested. Thank you. 

Discussion

On page 14 line 282, rewrite like this, “in this study, the rate of latrine facility utilization…”. Line 283 and 284 like this, “elsewhere in other parts of Ethiopia such as...”. Line 285 through 290 is too long sentence. Please keep it succinct and to the point.

Response: Thank you and we did as suggested (please check the revised version in lines 323 to 337). 

Try to rewrite a sentence from line 292 through 295 like this, After adjusting for all variables, a socio-demographic factors in a form of households with small family size of 1-3 members were significantly associated with latrine facility utilization as opposed to households with larger family members (4 person or more), which is consistent with studies in southeastern zone of Tigray(), and in Hawassa ( ). 

Response: We rewrite as suggested and please check in lines 339 to 344. 

Lines 297 through 299 is not clear, rewrite this paragraph.

Response: We re-write lines 297 to 299. 

Conclusion

In conclusion, there is no need to make discussion, just write the major conclusions and what recommendations they could made for policy makers. Your conclusion under abstract section is better stated than the one that was found at the last page of the manuscript. Move the concluding points under Abstract section to your Major “Conclusion” section in the last page of the manuscript. 

Response: We thank you for this key comment. We revised the conclusion as suggested (See the revised version of the conclusion). 

Acknowledgement

In PLoS ONE, you acknowledge funders not under Acknowledgment section but under funding section during online submission. 

Response: We deleted the funder in the acknowledgment section (See the revised version acknowledgment). 

Reference list

Change reference lists into Vancouver style please. Avoid capitalization problems as well. Do according to the Journal’s guideline.

Response: Yes, it is formatted as per the PLoS ONE guidelines for references. 

Overall, some editorial works are here in the manuscript. It should be made more concise and clearer in each section. They should check and revised thoroughly.

Response: Thank you for the remark and we revised the paper throughout. 

We would like to thank the reviewers and editors for evaluating our manuscript. We have tried to address all the concerns in a proper way and believe that our paper has been improved considerably. We would be happy to make further corrections if necessary and look forward to hearing from you all soon. 

I hope that the revised manuscript is accepted for publication in PLoS ONE. 

 Sincerely yours,

 Metadel Adane (PhD)

---

## [Decision Letter · Decision Letter 1]

13 Oct 2020

Household latrine utilization and associated factors in Semi-Urban areas of Northeastern Ethiopia

PONE-D-20-01859R1

Dear Dr. Adane (PhD),

We’re pleased to inform you that your manuscript has been judged scientifically suitable for publication and will be formally accepted for publication once it meets all outstanding technical requirements.

Kind regards,

Miguel A. Fernández, Ph.D.

Academic Editor

PLOS ONE

Additional Editor Comments (optional):

Reviewers' comments:

Reviewer's Responses to Questions

**Comments to the Author**

1. If the authors have adequately addressed your comments raised in a previous round of review and you feel that this manuscript is now acceptable for publication, you may indicate that here to bypass the “Comments to the Author” section, enter your conflict of interest statement in the “Confidential to Editor” section, and submit your "Accept" recommendation.

Reviewer #1: All comments have been addressed

Reviewer #2: All comments have been addressed

2. Is the manuscript technically sound, and do the data support the conclusions?

Reviewer #1: Yes

Reviewer #2: Yes

3. Has the statistical analysis been performed appropriately and rigorously? 

Reviewer #1: Yes

Reviewer #2: Yes

4. Have the authors made all data underlying the findings in their manuscript fully available?

Reviewer #1: Yes

Reviewer #2: Yes

5. Is the manuscript presented in an intelligible fashion and written in standard English?

Reviewer #1: Yes

Reviewer #2: Yes

6. Review Comments to the Author

Reviewer #1: It was nice to see the progress of the manuscript. The author has addressed all of the comments. It is now very well developed.

Reviewer #2: Reviewer # 1: Habtamu Tolera (PH.D)

The authors have revised the manuscript titled “Household Latrine utilization and its association with household family size in semi-urban areas of Alansha, south Wollo, northeastern, Ethiopia” as per the comments. The authors did the best job in the revision. I do not have more concerns and attachments in this review. Thus I recommend you to accept the paper without modification.

7. PLOS authors have the option to publish the peer review history of their article (what does this mean?). If published, this will include your full peer review and any attached files.

Reviewer #1: **Yes: **Mahfuza Islam

Reviewer #2: **Yes: **Habtamu Tolera

---

## [Editor Report · Acceptance letter]

4 Nov 2020

PONE-D-20-01859R1 

Household latrine utilization and associated factors in Semi-Urban areas of northeastern Ethiopia 

Dear Dr. Adane (PhD):

I'm pleased to inform you that your manuscript has been deemed suitable for publication in PLOS ONE. Congratulations! Your manuscript is now with our production department. 

Kind regards, 

on behalf of

Dr Miguel A. Fernández 

Academic Editor

PLOS ONE